# Enhanced accuracy and adaptability: An ISSA-optimized MPC approach for AGV trajectory tracking

**Tan Zhang, Chengjun Ding**[ID]*, **Tengfei Ma, Zijian Li, Zhikai Jing, Jinshen Yu, Jingyu Ge, Ping Duan, Jianing Zhang**

School of Mechanical Engineering, Hebei University of Technology, Tianjin, China

* dcj@hebut.edu.cn

## Abstract

To enhance the tracking accuracy of AGVs, this paper proposes an online adaptive optimization strategy for MPC weight parameters based on the Improved Sparrow Search Algorithm (ISSA). Firstly, the kinematic and three-degree-of-freedom dynamic models of the AGV are established, and a trajectory tracking MPC controller based on an incremental model is designed. On this basis, the Tent chaotic mapping is introduced to initialize the population and enhance its diversity. A dynamic disturbance factor is incorporated into the position update of the discoverers, and a Cauchy mutation operator is introduced during the follower stage. These improvements effectively balance the algorithm's global exploration and local exploitation capabilities, preventing premature convergence. This method uses a composite indicator of lateral and heading tracking errors as the fitness function to periodically optimize the MPC weight parameters online, thereby adapting to the control requirements of the AGV under different working conditions. Finally, validation through co-simulation using Gazebo and Rviz under the ROS framework, along with experiments on a forklift-type AGV, shows that the proposed method offers significant advantages in improving trajectory tracking accuracy, accelerating dynamic response, and enhancing adaptability to various working conditions. It thus provides a feasible solution for high-performance trajectory tracking control of AGVs.

## Introduction

In recent years, with the rapid advancement of technologies such as artificial intelligence and the Internet of Things, the manufacturing and logistics industries are accelerating their transformation towards intelligence and flexibility. As the core execution unit of intelligent logistics systems, Automated Guided Vehicles (AGVs) are playing an increasingly vital role in scenarios such as warehouse management [1–3], workshop material handling, and port container

**Data availability statement:** The relevant data are included in the manuscript and its Supporting Information files. The data related to the figures and tables have been uploaded to the system under the file name: Figures_and_Related_Data.zip.

**Funding:** This work was supported by the Tianjin Science and Technology Plan Project (Grant No. 25ZXRGGX00310), the Shijiazhuang Science and Technology Cooperation Special Project (Grant No. SJZZXA25003), and the Shijiazhuang Science and Technology Cooperation Special Project (Grant No. SJZZXB24002).

**Competing interests:** The authors have declared that no competing interests exist.

transportation, owing to their high precision, strong coordination capability, and adaptability. Particularly driven by "Industry 4.0," AGVs are now required not only to accomplish point-to-point transportation tasks [4,5] but also to achieve high-precision trajectory tracking [6] and multi-vehicle cooperative operations in dynamic and complex environments, which places higher demands on the control performance of AGVs.

Trajectory tracking control [7–9] is a key technology for AGVs to perform tasks such as transportation, docking, loading, and unloading. Its performance directly affects operational accuracy, system efficiency, and operational safety. Currently, mainstream path tracking control algorithms mainly include PID control [10], pure pursuit control [11], model predictive control (MPC) [12], and sliding mode control [13]. MPC demonstrates significant advantages in AGV trajectory tracking control due to its ability to explicitly handle system constraints and perform forward rolling optimization based on a model. However, the control performance of MPC highly depends on the appropriate setting of weight parameters in its cost function [14–16]. Traditional parameter tuning often relies on manual experience or trial-and-error methods. Under complex working conditions such as AGV load variations, sudden path curvature changes, and alterations in ground adhesion conditions, a fixed parameter set can hardly maintain optimal performance consistently. This may lead to decreased tracking accuracy, abrupt control actions, or even instability.

The forklift-type AGV adopts a unique mechanical structure design [17]. Its steering system utilizes active steering wheels to simultaneously achieve driving and steering functions, while coordinating with the two passive wheels located at the fork position [18–23]. From the perspective of kinematic characteristics, this type of intelligent handling equipment falls under the category of car-like mobile robots. Its dynamic model exhibits complex system features such as multiple inputs, underactuation, and nonholonomic constraints. To enhance the adaptability of MPC, researchers have conducted extensive studies focusing on two main directions: model-based adaptation and parameter-tuning-based adaptation. Model-based approaches include linear time-varying MPC [24], adaptive learning MPC, polytopic model-based robust predictive control [25], and human–machine robust shared control considering network delays [26]. These methods improve robustness to certain uncertainties but often rely on accurate system models and do not directly address the adaptive tuning of MPC weight parameters. Parameter-tuning-based approaches include fuzzy control [27,28], Bayesian optimization [29], and metaheuristic algorithms such as Particle Swarm Optimization (PSO) [30] and Ant Colony Optimization [31–34]. While metaheuristic-based methods have been explored for offline MPC parameter optimization, they typically lack online adaptation capability and often suffer from poor balance between global exploration and local exploitation, as well as premature convergence, which limits their effectiveness under dynamically changing operating conditions.

In recent years, several advanced approaches have emerged for MPC parameter tuning and predictive optimization, including Reinforcement Learning (RL) [35], event-triggered hybrid optimization [36], and the Alternating Direction Method of Multipliers (ADMM) [37]. RL-based methods leverage learned policies to adapt control

parameters online but often require extensive training data and may lack convergence guarantees in safety-critical applications. Event-triggered hybrid optimization reduces computational burden by performing optimization only when necessary, yet its performance heavily depends on the design of triggering thresholds. ADMM offers efficient distributed optimization but typically assumes convex problem structures, which may not hold in nonlinear AGV tracking scenarios. While these methods have demonstrated effectiveness in various control applications, they often involve significant computational complexity or rely on specific problem structures that are not always satisfied in the context of forklift-type AGV trajectory tracking under real-time constraints.

Overall, existing methods have not yet properly resolved the balance among global search capability, adaptability, and real-time performance. Especially in scenarios where forklift-type AGVs operate under varying conditions and require high control response, there is still a need for further development of efficient and reliable parameter adaptive optimization strategies [38–41].

To address the aforementioned issues, this paper proposes an online optimization strategy for MPC weight parameters in forklift-type AGV trajectory tracking based on an Improved Sparrow Search Algorithm (ISSA). Unlike existing offline metaheuristic-based tuning methods, our approach performs periodic online optimization triggered either at fixed intervals or by significant tracking errors, enabling real-time adaptation to changing operating conditions. In contrast to Reinforcement Learning (RL) and ADMM, ISSA-MPC requires no pre-training and does not rely on convex problem structures, while offering more predictable computational loads than event-triggered hybrid optimization. Three complementary mechanisms are introduced to overcome the limitations of standard SSA: (1) Tent chaotic mapping for enhanced population diversity; (2) a dynamic disturbance factor in the discoverer stage to balance global exploration and local exploitation; and (3) Cauchy mutation in the follower stage to escape local optima. A fitness function based on comprehensive tracking error is designed to periodically optimize the MPC weight parameters online, allowing the controller to adapt to different operating conditions of the forklift-type AGV. Finally, the effectiveness and superiority of the proposed method are validated through co-simulation using Gazebo and Rviz under the ROS architecture, as well as real-vehicle experiments.

The paper is structured as follows. Section 1 presents the vehicle kinematic and dynamic models. Section 2 details the design of the Model Predictive Controller. Section 3 proposes the adaptive time-domain MPC with an improved sparrow optimization algorithm. Section 4 validates the method through simulations and real-vehicle tests. Finally, Section 5 summarizes the conclusions.

## Establishment of the AGV motion model

**Kinematic model.** The Kinematic model serves as the foundation for implementing the AGV control system [42,43]. This paper takes a single-steering-wheel driven forklift-type AGV as an example, in which both steering and driving functions are integrated into the same steering wheel. When the forklift-type AGV travels with its forks leading, the vehicle body exhibits a larger swinging amplitude. To ensure tracking accuracy at the fork end for more precise insertion and retrieval operations, the midpoint of the axis connecting the two passive wheels is selected as the origin. By receiving positioning data from the radar and performing coordinate transformation, the coordinates of the vehicle's reference point are obtained. Fig 1 illustrates this kinematic model.

In Fig 1, {XOY} represents the world coordinate system, and {xoy} denotes the coordinate system of the forklift-type AGV. $(x_f, y_f)$ and $(x_r, y_r)$ correspond to the coordinates of the front wheel reference point and the rear axle center point, respectively; $v_r$ is the velocity of the rear axle reference point; $l$ is the wheelbase of the forklift-type AGV; $R$ is the turning radius of the forklift-type AGV; $v_f$ is the traveling speed of the steering wheel; $\delta_f$ represents the steering angle of the steering wheel; $\varphi$ is the heading angle of the forklift-type AGV.

$$v_r = \dot{X}_r \cos\varphi + \dot{Y}_r \sin\varphi \tag{1}$$

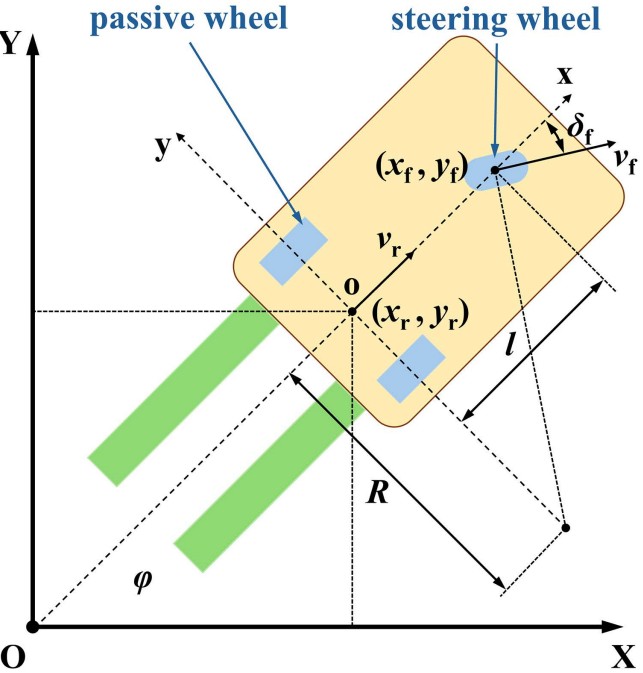

**Fig 1. Kinematic Model of the Forklift-type AGV**At the center point of the rear axle of the forklift-type AGV, the velocity is:.

The kinematic constraints for the front wheel and the rear axle are:

$$\begin{cases} \dot{X}_f \sin\left(\varphi + \delta_f\right) - \dot{Y}_f \cos\left(\varphi + \delta_f\right) = 0 \\ \dot{X}_r \sin\varphi - \dot{Y}_r \cos\varphi = 0 \end{cases}$$

(2)

Combining Equation (1) and Equation (2) yields:

$$\begin{cases} \dot{X}_r = v_r \cos\varphi \\ \dot{Y}_r = v_r \sin\varphi \end{cases}$$

(3)

From the geometric relationships in Fig 1, we obtain:

$$\begin{cases} X_f = X_r + l \cos\varphi \\ Y_f = Y_r + l \sin\varphi \end{cases}$$

(4)

By simultaneously substituting Equation (3) and Equation (4) into Equation (2), the angular velocity of the heading angle is derived as:

$$\omega = \left(v_r/l\right) \tan \delta_f$$

(5)

Based on $\omega$ and $v_r$, the turning radius $R$ and the steering wheel angle $\delta_f$ are further solved as:

$$\begin{cases} R = v_r/\omega \\ \delta_f = \arcsin\left(l/R\right) \end{cases}$$

(6)

By integrating Equation (3) and Equation (5), the kinematic model of the forklift-type AGV is obtained as:

$$\left[\dot{X}_r, \dot{Y}_r, \dot{\varphi}\right]^T = \left[\cos\varphi, \sin\varphi, \tan\delta_f/l\right]^T v_r \tag{7}$$

**Dynamic model.** The kinematic model simplifies the AGV as a mass point, considering only its kinematic characteristics such as velocity and angle, while ignoring the potential disturbances to tracking control caused by vehicle force conditions [44]. However, during operation, the AGV's speed, ground contact area, and forces vary across different operational phases, which affects tracking performance to some extent.

To further enhance trajectory tracking performance, dynamic modeling of the AGV is required. To balance model accuracy and computational efficiency, the dynamic model of the forklift-type AGV is appropriately simplified in this paper, and modeling is conducted based on the following reasonable assumptions:

(1) The entire forklift-type AGV is considered as a rigid system;

(2) It operates only on a planar surface, neglecting vertical motion;

(3) The effects of lateral load transfer and longitudinal-lateral coupling of the steering wheel on the motion of the forklift-type AGV are ignored;

(4) The influence of aerodynamics is neglected.

Based on the aforementioned assumptions, a three-degree-of-freedom vehicle dynamic model is established, as shown in Fig 2 For computational convenience, the two passive wheels on the rear axle are simplified as a single wheel, collinear with the center of the steering wheel. In Fig 2, {XOY} represents the world coordinate system, and {xoy} denotes the vehicle coordinate system; $F_{yf}$ and $F_{yr}$ are the lateral forces of the front and rear wheels, respectively, while $F_{xf}$ and $F_{xr}$ are the corresponding longitudinal forces; $\varphi$ is the heading angle; $\dot{\varphi}$ is the angular velocity of the heading angle; $\beta$ is the sideslip angle at the center of mass; $v_x$ and $v_y$ represent the longitudinal and lateral velocities,

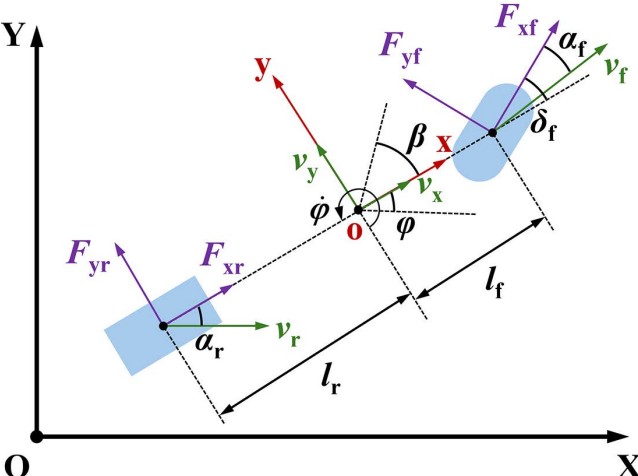

**Fig 2. Three-Degree-of-Freedom AGV Dynamic Model.** By conducting a force analysis on the longitudinal, lateral, and yaw dynamics of the forklift-type AGV, we obtain.

respectively; $l_f$ and $l_r$ are the distances from the center of mass to the front and rear axles; $\alpha_f$ and $\alpha_r$ denote the slip angles of the front and rear wheels.

$$\begin{cases} m\left(\dot{v}_x - v_y\dot{\varphi}\right) = F_{xf}\cos\delta_f - F_{yf}\sin\delta_f + F_{xr} \\ m\left(\dot{v}_y + v_x\dot{\varphi}\right) = F_{xf}\sin\delta_f + F_{yf}\cos\delta_f + F_{yr} \\ I_z\ddot{\varphi} = l_f\left(F_{xf}\sin\delta_f + F_{yf}\cos\delta_f\right) - l_r F_{yr} \end{cases}$$

(8)

where $m$ is the vehicle mass, and $I_z$ is the moment of inertia about the z-axis of the {xoy} coordinate system.

When the wheel slip angles and slip ratios are small, the wheel forces derived from the Magic Formula can be approximated by a linear relationship:

$$\begin{cases} F_{xf} = C_{lf}S_f, F_{yf} = C_{\alpha f}\alpha_f \\ F_{xr} = 2C_{lr}S_r, F_{yr} = 2C_{\alpha r}\alpha_r \end{cases}$$

(9)

where $C_{lf}$ and $C_{lr}$ represent the longitudinal stiffness of the front and rear wheels, respectively, with the corresponding cornering stiffnesses denoted as $C_{\alpha f}$ and $C_{\alpha r}$; $S_f$ and $S_r$ are the slip ratios of the front and rear wheels, respectively.

Using a linear tire model with the small-angle assumption yields:

$$\begin{cases} \alpha_f = \arctan\left(\frac{v_y + l_f\dot{\varphi}}{v_x}\right) - \delta_f \approx \frac{v_y + l_f\dot{\varphi}}{v_x} - \delta_f \\ \alpha_r = \arctan\left(\frac{v_y - l_f\dot{\varphi}}{v_x}\right) \approx \frac{v_y - l_r\dot{\varphi}}{v_x} \end{cases}$$

(10)

To reduce computational complexity, the AGV dynamic model in Equation (10) assumes that both the front-wheel steering angle and the wheel slip angles remain within a small range. Consequently, the trigonometric functions can be simplified as:

$$\cos\theta \approx 1, \sin\theta \approx \theta, \tan\theta \approx \theta$$

(11)

To characterize the AGV's motion in the global frame, the coordinate transformation between the vehicle-fixed frame and the global frame is derived.

$$\begin{cases} \dot{Y} = v_x\sin\varphi + v_y\cos\varphi \\ \dot{X} = v_x\cos\varphi - v_y\sin\varphi \end{cases}$$

(12)

Integrating Equation (9) to Equation (12) leads to the final three-degree-of-freedom AGV dynamic model, which incorporates the Magic Formula and a small-angle assumption:

$$\begin{cases} \dot{v}_y = -v_x\dot{\varphi} + \frac{1}{m}\left[C_{lf}S_f\delta_f + C_{\alpha f}\left(\delta_f - \frac{v_y + l_f\dot{\varphi}}{v_x}\right) + 2C_{\alpha r}\frac{l_r\dot{\varphi} - v_y}{v_x}\right] \\ \dot{v}_x = v_y\dot{\varphi} + \frac{1}{m}\left[C_{lf}S_f + C_{\alpha f}\left(\delta_f - \frac{v_y + l_f\dot{\varphi}}{v_x}\right)\delta_f + 2C_{lr}S_r\right] \\ \ddot{\varphi} = \frac{1}{I_z}\left[l_f C_{\alpha f}\left(\delta_f - \frac{v_y + l_f\dot{\varphi}}{v_x}\right) - 2l_r C_{\alpha r}\frac{l_r\dot{\varphi} - v_y}{v_x}\right] \\ \dot{Y} = v_x\sin\varphi + v_y\cos\varphi \\ \dot{X} = v_x\cos\varphi - v_y\sin\varphi \end{cases}$$

(13)

It should be noted that the vehicle dynamics model in this paper is derived based on several simplifying assumptions, including small slip angles, planar motion, and simplified tire force relationships. Under operating conditions such as sharp turns or significant load variations, the actual vehicle behavior may deviate from these assumptions, potentially leading

to model–plant mismatches. Therefore, it is particularly important to clarify the applicable operating range of the linear tire model assumptions. In this study, the maximum speed of the forklift-type AGV is approximately 1.5 m/s, which aligns with typical factory floor operations. During indoor navigation, the speed is generally limited to within 1 m/s. Within this speed range, the wheel slip angles typically remain below 3–5°, which falls within the linear region of typical tire cornering stiffness characteristics. Consequently, the linear tire model provides sufficient tracking control accuracy under the tested conditions. In addition, the proposed ISSA-MPC framework offers inherent robustness against such modeling inaccuracies. By periodically optimizing the MPC weight parameters online based on real-time tracking errors, the controller can adapt to unmodeled dynamics without requiring an explicit model update.

**MPC controller design**

**Linearization and discretization of nonlinear systems.** Based on Equation (13), the nonlinear AGV dynamic state-space equation is established as:

$$\begin{cases} \dot{\xi}(t) = f(\xi(t), \boldsymbol{u}(t)) \\ \eta(t) = \boldsymbol{C}\xi(t) \end{cases}$$

(14)

where the state vector is $\xi = [\dot{y}, \dot{x}, \varphi, \dot{\varphi}, Y, X]^T$, the control input vector is $\boldsymbol{u} = [\delta_f]$, the output vector is $\eta = [\varphi, Y]^T$, and $\boldsymbol{C}$ is the output matrix, $\boldsymbol{C} = \begin{bmatrix} 0 & 0 & 1 & 0 & 0 & 0 \\ 0 & 0 & 0 & 0 & 1 & 0 \end{bmatrix}$.

Due to the high real-time requirements for AGV motion control, linearization is performed. Assuming the control inputs remain unchanged, a Taylor series expansion is applied at the operating point $(\xi_0, \boldsymbol{u}_0)$. By neglecting the higher-order terms and retaining only the first-order terms, it can be expressed as:

$$\begin{aligned} \xi(t+1) &= f(\xi_0(t), \boldsymbol{u}_0(t)) + \boldsymbol{A}(t)(\xi(t) - \xi_0(t)) \\ &+ \boldsymbol{B}(t)(\boldsymbol{u}(t) - \boldsymbol{u}_0(t)) \end{aligned}$$

(15)

where $\boldsymbol{A}(t) = \frac{\partial f(\xi_0, \boldsymbol{u}_0)}{\partial \xi}$, and $\boldsymbol{B}(t) = \frac{\partial f(\xi_0, \boldsymbol{u}_0)}{\partial \boldsymbol{u}}$.

The system is discretized using the forward Euler method. After rearrangement, the discrete-time state equation is obtained as:

$$\begin{cases} \xi(k+1) = \boldsymbol{A}_{k,t}\xi(k) + \boldsymbol{B}_{k,t}\boldsymbol{u}(k) + \boldsymbol{d}_{k,t} \\ \eta(k) = \boldsymbol{C}_{k,t}\xi(k) \end{cases}$$

(16)

where $\boldsymbol{A}_{k,t} = \boldsymbol{I} + \boldsymbol{T}\boldsymbol{A}(t)$, $\boldsymbol{B}_{k,t} = \boldsymbol{T}\boldsymbol{B}(t)$, $\boldsymbol{I}$ is the identity matrix, $\boldsymbol{d}_{k,t} = \xi_t(k+1) - \boldsymbol{A}_{k,t}\xi_t(k) - \boldsymbol{B}_{k,t}\xi_t(k)$, $\boldsymbol{C}_{k,t}$ is the output matrix, $\boldsymbol{T}$ is the sampling period, and $k$ denotes any discrete time step.

**Output prediction equations**

To achieve smooth control of the forklift-type AGV, an incremental formulation is introduced, where the control input in Equation (16) is reformulated in terms of its increment:

$$\Delta\boldsymbol{u}(k|t) = \boldsymbol{u}(k|t) - \boldsymbol{u}(k-1|t)$$

(17)

By defining $\tilde{\xi}(k+1|t) = [\xi(k+1|t), u(k|t)]^T$, a new state-space equation is derived:

$$\begin{cases} \tilde{\xi}(k+1|t) = \tilde{\boldsymbol{A}}_{k,t}\tilde{\xi}(k|t) + \tilde{\boldsymbol{B}}_{k,t}\Delta\boldsymbol{u}(k|t) + \tilde{\boldsymbol{d}}_{k,t} \\ \eta(k|t) = \tilde{\boldsymbol{C}}_{k,t}\tilde{\xi}(k|t) \end{cases}$$

(18)

where $\tilde{A}_{k,t} = \begin{bmatrix} A_{k,t} & B_{k,t} \\ 0_{1\times 6} & I \end{bmatrix}$, $\tilde{B}_{k,t} = \begin{bmatrix} B_{k,t} \\ I \end{bmatrix}$, $\tilde{d}_{k,t} = \begin{bmatrix} d_{k,t} \\ 0 \end{bmatrix}$, $\tilde{C}_{k,t} = \begin{bmatrix} C_{k,t} & 0 \end{bmatrix}$.

To further reduce the controller's complexity and accelerate the solution process, the following assumptions are proposed:

$$\begin{cases} \tilde{A}_{k,t} = \tilde{A}_t, k = 1, \cdots, t+N-1 \\ \tilde{B}_{k,t} = \tilde{B}_t, k = 1, \cdots, t+N-1 \\ \tilde{C}_{k,t} = \tilde{C}_t, k = 1, \cdots, t+N-1 \end{cases} \tag{19}$$

Based on the fundamental principle of MPC, the prediction horizon and control horizon are defined as $N_p$ and $N_c$, respectively, and the following assumptions are made:

$$\Delta u\left(k + N_c \mid t\right) = \Delta u\left(k + N_c + 1 \mid t\right) = \cdots = \Delta u\left(k + N_p - 1 \mid t\right) = 0 \tag{20}$$

At a given time step $k$, predictions for the state variables and system outputs are made over the future prediction horizon $N_p$. The sequences of states and outputs within the prediction horizon $N_p$ are given by:

$$\begin{cases} \tilde{\xi}\left(k + N_p \mid t\right) = \tilde{A}_t^{N_p} \tilde{\xi}\left(k \mid t\right) + \tilde{A}_t^{N_p-1} \tilde{B}_t \Delta u\left(k \mid t\right) + \\ \quad \cdots + \tilde{A}_t^{N_p-N_c-1} \tilde{B}_t \Delta u\left(k + N_c \mid t\right) + \tilde{d}_{k,t} \\ \eta\left(k + N_p \mid t\right) = \tilde{C}_t \tilde{A}_t^{N_p} \tilde{\xi}\left(k \mid t\right) + \tilde{C}_t \tilde{B}_t \Delta u\left(k \mid t\right) + \\ \quad \cdots + \tilde{C}_t \tilde{A}_t^{N_p-N_c-1} \tilde{B}_t \Delta u\left(k + N_c \mid t\right) \end{cases} \tag{21}$$

The output prediction equations of the system are expressed in matrix form as:

$$Y\left(t\right) = \Psi_t \xi\left(t \mid t\right) + \Theta_t \Delta U\left(t\right) + \Gamma_t \Phi\left(t\right) \tag{22}$$

where $Y\left(k + 1\right)$, $\xi\left(k\right)$, and $\Phi\left(k\right)$ represent the output sequence, state sequence, and error sequence over the prediction horizon $N_p$, respectively; $\Delta U\left(k\right)$ denotes the control increment sequence over the control horizon $N_c$; and $\Psi_k$, $\Theta_k$, and $\Gamma_k$ are coefficient matrices.

## Objective function design

The core objective of trajectory tracking is to minimize the tracking deviation of the forklift-type AGV while ensuring tracking stability [45–49]. The cost function must incorporate both the vehicle's output variables and control increments, and is designed as follows:

$$J_{\min}\left(\xi\left(t\right), u\left(t-1\right), \Delta U\left(t\right), \varepsilon\right) = \sum_{i=1}^{N_p} \left\| \eta\left(t+i \mid t\right) - \eta_{\text{ref}}\left(t+i \mid t\right) \right\|_Q^2$$
$$+ \sum_{i=1}^{N_c-1} \left\| \Delta u\left(t+i \mid t\right) \right\|_R^2 + \rho\varepsilon^2 \tag{23}$$

where $Q$ and $R$ denote weight matrices; $\rho$ is the weighting coefficient for the slack variable; $\varepsilon$ represents the slack variable; and $\eta_{\text{ref}}$ is the reference value for the output.

## Reformulation as a quadratic programming problem for solution

To facilitate the controller's programming and solution, Equation (23) is transformed into the following form after applying assumptions and derivations, while neglecting constant terms that do not affect the optimization objective:

$$J_{\min} \left( \boldsymbol{\xi} \left( t \right), \boldsymbol{u} \left( t-1 \right), \Delta \boldsymbol{U} \left( t \right), \varepsilon \right) = \begin{bmatrix} \Delta \boldsymbol{U} \left( t \right) \\ \varepsilon \end{bmatrix}^{\mathrm{T}} \boldsymbol{H}_t \begin{bmatrix} \Delta \boldsymbol{U} \left( t \right) \\ \varepsilon \end{bmatrix} + \boldsymbol{G}_t \begin{bmatrix} \Delta \boldsymbol{U} \left( t \right) \\ \varepsilon \end{bmatrix} \tag{24}$$

where $\boldsymbol{H}_t$ is a positive definite Hessian matrix with $\boldsymbol{H}_t = \begin{bmatrix} \Theta_t^{\mathrm{T}} \boldsymbol{Q} \Theta_t + \boldsymbol{R} & \boldsymbol{0} \\ \boldsymbol{0} & \rho \end{bmatrix}$, and $\boldsymbol{G}_t = \begin{bmatrix} 2\boldsymbol{E}_t^{\mathrm{T}} \boldsymbol{Q} \Theta_t & \boldsymbol{0} \end{bmatrix}$.

After comprehensively considering the objective function and constraints, the controller needs to solve the following problem in each control cycle:

$$J_{\min} \left( \boldsymbol{\xi} \left( t \right), \boldsymbol{u} \left( t-1 \right), \Delta \boldsymbol{U} \left( t \right), \varepsilon \right) = \text{s.t.} \begin{cases} \Delta \boldsymbol{U}_{\min} \leq \Delta \boldsymbol{U} \left( t \right) \leq \Delta \boldsymbol{U}_{\max} \\ \boldsymbol{U}_{\min} \leq \boldsymbol{U} \left( t \right) \leq \boldsymbol{U}_{\max} \\ \boldsymbol{y}_{h,\min} \leq \boldsymbol{y}_h \leq \boldsymbol{y}_{h,\max} \\ \boldsymbol{y}_{s,\min} - \varepsilon \leq \boldsymbol{y}_s \leq \boldsymbol{y}_{s,\max} + \varepsilon \end{cases} \tag{25}$$

where $\boldsymbol{U}_{\max}$ and $\boldsymbol{U}_{\min}$ are the upper and lower bounds of the control inputs, respectively; $\Delta \boldsymbol{U}_{\max}$ and $\Delta \boldsymbol{U}_{\min}$ are the upper and lower bounds of the control increments, respectively; $\boldsymbol{y}_h$, and $\boldsymbol{y}_s$ represent the hard-constrained outputs and soft-constrained outputs, respectively.

### Feedback correction

The control increment sequence to be solved in each control cycle is:

$$\Delta \boldsymbol{U} \left( t \right) = \begin{bmatrix} \Delta \boldsymbol{u} \left( t \right), \Delta \boldsymbol{u} \left( t+1 \right), \cdots, \Delta \boldsymbol{u} \left( t + N_c - 1 \right) \end{bmatrix} \tag{26}$$

Applying the first element of the sequence to the system yields the actual control increment:

$$\boldsymbol{u} \left( t \right) = \boldsymbol{u} \left( t-1 \right) + \Delta \boldsymbol{u} \left( t \right) \tag{27}$$

The system continuously generates new control sequences through receding horizon optimization until the trajectory tracking control objective is achieved.

### Adaptive time-domain MPC controller design

**Impact of time-domain parameters on tracking performance.** The values of the prediction horizon $N_p$ and the control horizon $N_c$ in the MPC controller both significantly impact the tracking control performance. When other parameters of the MPC controller remain unchanged, a larger $N_p$ can cover a longer trajectory, but model errors accumulate over the prediction steps, reducing the reliability of long-term predictions and potentially increasing tracking errors. Conversely, a smaller $N_p$ may fail to anticipate future trajectory changes, leading to tracking lag or overshoot. Furthermore, with other controller parameters fixed, increasing $N_c$ allows more flexible multi-step adjustments, which theoretically enables more accurate trajectory tracking. However, due to the physical limitations of the actuator, the actual control inputs may not reach the optimized solution, potentially causing saturation or instability and thereby reducing real-time performance. Decreasing $N_c$ can improve stability and real-time performance but may result in untimely correction of tracking errors, thus reducing tracking accuracy.

Therefore, the selection of time-domain parameters requires a comprehensive consideration of factors such as tracking accuracy, stability, and real-time performance [50,51]. The time-domain parameters of an MPC controller are often chosen based on debugging experience, making it difficult to adapt to dynamic working condition changes, which can lead to reduced tracking accuracy or fluctuating control inputs. Traditional parameter adjustment methods—such as manual trial-and-error and basic grid search—suffer from poor real-time performance when applied to online MPC weight

adaptation, as they lack efficient search mechanisms [15,31]. Meanwhile, conventional metaheuristic algorithms used for offline tuning are prone to premature convergence to local optima in complex, high-dimensional search spaces [30,31]. Hence, intelligent algorithms with enhanced global search capability are frequently employed to optimize these parameters. The Sparrow Search Algorithm (SSA) has gained considerable attention due to its simple structure and ease of implementation, but it is also prone to converging to local optimal solutions. To address this, this paper improves the Sparrow Search Algorithm and proposes an Improved Sparrow Search Algorithm (ISSA) to identify the optimal time-domain combination for the forklift-type AGV under varying working conditions, and to periodically optimize the MPC weight parameters to adapt to different operating conditions. In our implementation, the ISSA optimization is triggered every 50 control cycles (at a control frequency of 25 Hz, i.e., approximately every 2 seconds). Additionally, when the system detects a significant deviation from the reference trajectory (e.g., lateral error > 0.05 m for 0.5 s), the optimization is activated earlier to promptly respond to sudden changes in operating conditions.

The improvements include: introducing Tent mapping to initialize the sparrow population, thereby enhancing the randomness of population distribution; employing a nonlinear dynamic adjustment strategy for the inertia weight and learning factors of the sparrow population to further optimize the algorithm's search capability; and applying random perturbations to sparrow positions by integrating quasi-reflection learning and Gaussian mutation, combined with a greedy mechanism to prevent the algorithm from becoming trapped in local optima.

## Improved sparrow search algorithm

The traditional Sparrow Search Algorithm simulates the foraging and anti-predation behaviors of sparrow flocks through a three-stage mechanism involving "discoverers, followers, and sentinels." SSA is known to suffer from three main limitations when applied to complex optimization problems such as MPC parameter tuning: (1) random initialization often leads to uneven population distribution and insufficient global exploration; (2) the balance between global exploration and local exploitation is difficult to maintain, especially in high-dimensional search spaces; and (3) the algorithm is prone to premature convergence to local optima. To address these limitations, we introduce three complementary mechanisms—Tent chaotic initialization, a dynamic disturbance factor for discoverers, and Cauchy mutation for followers—each specifically tailored to enhance the performance of SSA in the context of online MPC weight adaptation.

**Discovery phase with dynamically adjusted exploration and exploitation.** The discoverers are responsible for global exploration, guiding the population to explore new solutions under the lead of the best individuals, thereby avoiding local optima. The top 20% of the fittest individuals are selected as "discoverers." A random perturbation is applied to the weight parameter of each discoverer to expand the search scope. The position update formula for the discoverers is:

$$X_{i,j}^{t-1} = \begin{cases} X_{i,j}^t \cdot \exp\left(-\frac{i}{\alpha T}\right) + QL, & \text{if discoverer position superior} \\ X_{i,j}^t + K\left(X_{best}^t - X_{i,j}^t\right) + \delta \cdot \text{randn}(\ ), & \text{otherwise} \end{cases} \tag{28}$$

where $X_{i,j}^t$ denotes the position of the $i$-th sparrow in the $j$-th dimension at generation t; $a$ is a random number within [0,1]; $T$ is the maximum number of iterations; $Q$ is a normally distributed random number; $L$ is an all-ones vector; $K$ is a random number in [0.5,1], controlling the proximity of followers to discoverers; $\delta(\delta \in [0, 0.1])$ is a disturbance factor used to enhance global exploration capability; and $X_{best}$ represents the current optimal position.

The dynamic disturbance factor is designed to address the exploration–exploitation trade-off. In MPC parameter tuning, the optimal prediction horizon $N_p$ and control horizon $N_c$ may vary significantly under different path curvatures or vehicle speeds. Early-stage exploration is crucial to identify promising regions, while late-stage fine-tuning ensures precise tracking performance. The decreasing disturbance factor enables discoverers to perform coarse exploration initially and gradually shift to refined local search, preventing the algorithm from either being trapped prematurely or converging too slowly.

**Adaptive mutation in the follower phase.** The remaining 80% of individuals act as followers, leveraging the current optimal information to conduct local refinement search. Each follower randomly selects a discoverer as its leader. Followers optimize their own positions by imitating the discoverers. However, the traditional algorithm easily leads to a decline in population diversity. To address this, the present invention introduces a Cauchy mutation operator to improve the position update formula for the followers:

$$X_{i,j}^{t+1} = X_{leader,j}^t + \beta \cdot |X_{i,j}^t - X_{best}^t| + \gamma \cdot Cauchy\,(0, \sigma) \tag{29}$$

where $X_{leader,j}^t$ denotes the $j$-th dimensional position of the leader sparrow for the $i$-th follower; $\beta$ is a random number within [0.3,0.7]; $\gamma$ is the mutation intensity coefficient, given by $\gamma = 0.1(1-t/T)$; Cauchy(0, $\sigma$) is a random number following the standard Cauchy distribution (with $\sigma = 0.5$), used to escape local optima.

Cauchy mutation is introduced to mitigate the problem of premature convergence, which is particularly detrimental in MPC parameter tuning because suboptimal weights can cause unstable control behavior. Unlike Gaussian mutation, the Cauchy distribution has a heavier tail, generating larger perturbations with higher probability. This allows followers to escape local optima more effectively while the greedy selection mechanism ensures that only beneficial mutations are retained. This mechanism is essential for maintaining tracking performance under changing operating conditions.

**Dynamic boundary constraints in the sentinel phase.** Sentinels are responsible for monitoring population safety and trigger a random position reset upon detecting danger. Whereas the traditional algorithm simply truncates solutions that exceed boundaries—which often causes the population to cluster near the boundaries—this paper adopts a combined reflective boundary strategy and random reset strategy:

$$X_{i,j}^{t+1} = \begin{cases} 2L_j - X_{i,j}^t, & \text{if} X_{i,j}^t > U_j, \text{ reflection correction} \\ 2U_j - X_{i,j}^t, & \text{if} X_{i,j}^t < U_j, \text{ reflection correction} \\ X_{best}^t + \eta \cdot \text{randn}\,(\,), & \text{if} C_{reset} \end{cases} \tag{30}$$

where $L_j$ and $U_j$ are the lower and upper bounds of the variable in the $j$-th dimension; $\eta$ is a random number within [0.1, 0.3]; and the condition $C_{reset}$ is the stagnation reset condition, which triggers a random reset if no improvement in fitness is observed for 5 consecutive generations.

The reflective boundary prevents the accumulation of the population near the boundaries, while the random reset strategy introduces new search directions into the population, thereby enhancing the global search capability.

The ISSA optimizer adjusts $N_p$ and $N_c$ jointly based on the current tracking error. When the vehicle approaches a high-curvature segment, the increasing lateral error triggers the selection of a moderately longer prediction horizon to improve future path anticipation, paired with a properly sized control horizon to enable smooth yet responsive steering. After passing the curve, the optimizer reverts to shorter horizons suited for straight-line tracking. This adaptive strategy directly addresses the compromise inherent in fixed-horizon MPC, allowing the controller to dynamically balance look-ahead capability and real-time responsiveness.

## Fitness function selection

The ISSA algorithm determines whether the current time-domain parameters are superior based on the fitness value. This paper selects the Integral of Time multiplied by Absolute Error (ITAE) index for lateral tracking error and heading angle error as the fitness function. This index comprehensively considers the impact of both vehicle tracking accuracy and stability. Its evaluation formula is given by:

$$J_{ITAE} = \lambda_1 \int_0^\infty t\,|Y_{error}\,(t)|\,dt + \lambda_2 \int_0^\infty t\,|\varphi_{error}\,(t)|\,dt \tag{31}$$

where $t$ is the current running time of the model; $Y_{error}(t)$ is the lateral tracking error at time $t$; $\varphi_{error}(t)$ is the heading angle error at time $t$; $\lambda_1$ and $\lambda_2$ are the weighting coefficients for each error term.

The ITAE-based fitness function evaluates the tracking performance under candidate weight parameters. Integration with MPC – The ISSA algorithm runs as a background process and is periodically executed according to the optimization triggering mechanism described above. At each optimization trigger, ISSA evaluates candidate weight parameters using the current vehicle state and a short preview of the reference trajectory. After convergence, the optimal weights are passed to the MPC controller. The controller then uses these updated weights in its cost function for subsequent control cycles until the next optimization is triggered or the control task ends. This mechanism ensures that the controller adapts to changing operating conditions while maintaining real-time performance.

### Tent chaotic map

The Sparrow Search Algorithm uses random values to initialize the population, which results in insufficient uniformity in the spatial distribution of the initial population. This leads to diminished search capability in the later stages and makes the algorithm prone to converging to local optima. To address this, a chaotic map is introduced to initialize the sparrow population, thereby enhancing its diversity. The point set generated by the Logistic map exhibits a distribution characteristic that is sparser in the middle and denser at the edges, as shown in Fig 3 This non-uniformity can adversely affect the final optimization accuracy of the algorithm. In contrast, the Tent map demonstrates a more uniform distribution in the space, as also illustrated in Fig 4 The piecewise linear structure of the Tent map leads to a more even sequence distribution, which helps improve the randomness and diversity of the initial population solutions.

$$z_{i+1} = \begin{cases} 2z_i + \text{rand}(0,1) \times \frac{1}{N_r}, & 0 \leq z_i \leq \frac{1}{2} \\ 2(1 - 2z_i) + \text{rand}(0,1) \times \frac{1}{N_r}, & \frac{1}{2} < z_i \leq 1 \end{cases}$$

(32)

where $N_r$ is the number of particles in the Tent chaotic sequence, and rand(0,1) is a random number between 0 and 1.

The flowchart of the improved ISSA is shown in Fig 5.

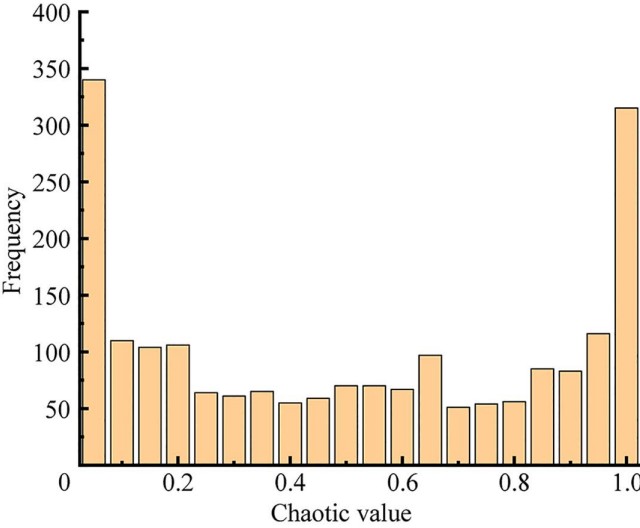

**Fig 3. Logistic map.** Point set distribution of Logistic map, showing sparser distribution in the middle and denser at the edges.

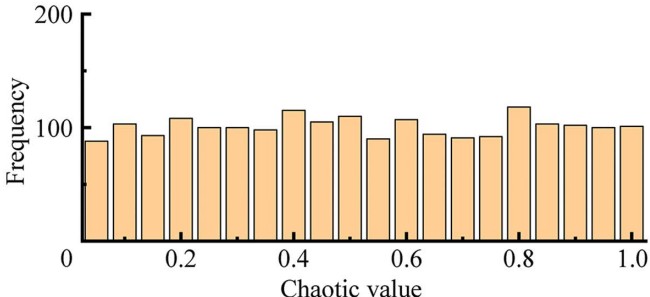

**Fig 4. Tent map.** Point set distribution of Tent map, demonstrating a more uniform spatial distribution.The Tent map is adopted to replace random values for initializing the sparrow population. Since the Tent map sequence exhibits small-scale periodicity, a random value is introduced into its fundamental expression to prevent the sequence from falling into periodic behavior. The improved expression is given as.

## Simulation analysis and experimental verification

This paper first validates the adaptive time-domain MPC controller through co-simulation using Gazebo and Rviz under the ROS framework, conducting a comparative analysis of the tracking accuracy and stability between the fixed time-domain MPC (FT-MPC) and the adaptive time-domain ISSA-MPC (AT-ISSA-MPC) controllers. The double lane change maneuver and the straight-line deviation correction scenario are selected for simulation. The simulation analysis is performed under the condition of a vehicle speed $v = 2$ m/s. The fixed time-domain MPC parameters are set as $N_p = 20$ and $N_c = 10$, while the time-domain parameters of the adaptive time-domain MPC are adjusted online according to the specific scenario. Finally, real-vehicle experiments are conducted using a forklift-type AGV experimental platform to verify the practicality of the proposed algorithm.

## Simulation analysis of the double lane change maneuver

The double lane change maneuver, originally developed for automotive handling tests, is adopted here as a representative scenario of continuous curvature variation. This allows rigorous evaluation of the controller's dynamic tracking performance under lateral maneuvers that mimic obstacle avoidance and aisle turning in warehouse environments.

Figs 6–9 present the simulation results of the two controllers under the double lane change simulation conditions. Figs 6 and 7 depict the lateral displacement and its corresponding error, respectively, while Figs 8 and 9 show the heading angle and its error, respectively.

Under the double lane change maneuver, both controllers can track the reference trajectory. As shown in Figs 7 and 9, the lateral and heading errors of both controllers peak near the two regions of maximum curvature. This is because high-curvature segments demand larger steering angles and lateral accelerations, posing greater challenges for trajectory tracking.

The FT-MPC uses constant weights throughout the maneuver. While this may perform adequately in low-curvature segments, it represents a compromise that cannot achieve optimal tracking across varying curvatures. In contrast, the AT-ISSA-MPC periodically optimizes weights online based on real-time tracking errors. When approaching a high-curvature segment, the increasing error triggers the optimizer to adjust weights thereby reducing the peak error. In addition to weight adaptation, the ISSA algorithm also adjusts the prediction horizon $N_p$ and control horizon $N_c$. At high-curvature segments, the optimizer selects a longer $N_p$ to anticipate the upcoming curve and an appropriately sized $N_c$ to allow smooth steering adjustments, thereby reducing the peak tracking error.

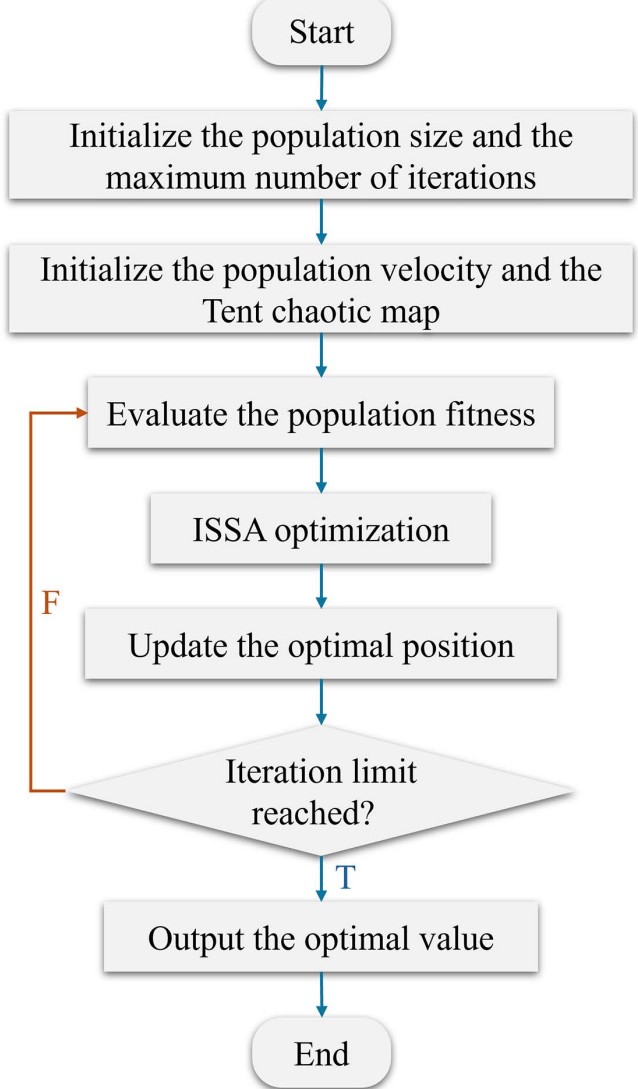

**Fig 5. ISSA flowchart.** In summary, The uniform distribution provided by Tent chaotic initialization is particularly advantageous for MPC parameter tuning, where the optimal weight combination may lie anywhere within a bounded search space. A well-distributed initial population increases the probability of covering the feasible region from the outset, reducing the risk of the algorithm converging to suboptimal parameters due to poor initialization. This directly addresses the limitation of the standard SSA, which often suffers from uneven population distribution and insufficient global exploration when optimizing MPC weight parameters.

As observed in Figs 7 and 9, the heading error peaks slightly after the lateral error. This order is physically plausible: when the vehicle enters a high-curvature region, lateral deviation develops first, followed by steering adjustments that subsequently induce heading deviation.

A comparative analysis using peak error and mean square error is presented in Tables 1 and 2. The AT-ISSA-MPC reduces the peak lateral error by 27.88% (from 0.09960 m to 0.07183 m) and its mean square error by 51.45% (from 0.001638 to 0.000795). The peak heading error is reduced by 36.54% (from 1.19386° to 0.75766°), and its mean square error by 59.72% (from 0.118311° to 0.047652°). These results demonstrate that AT-ISSA-MPC achieves higher tracking accuracy and stability than FT-MPC.

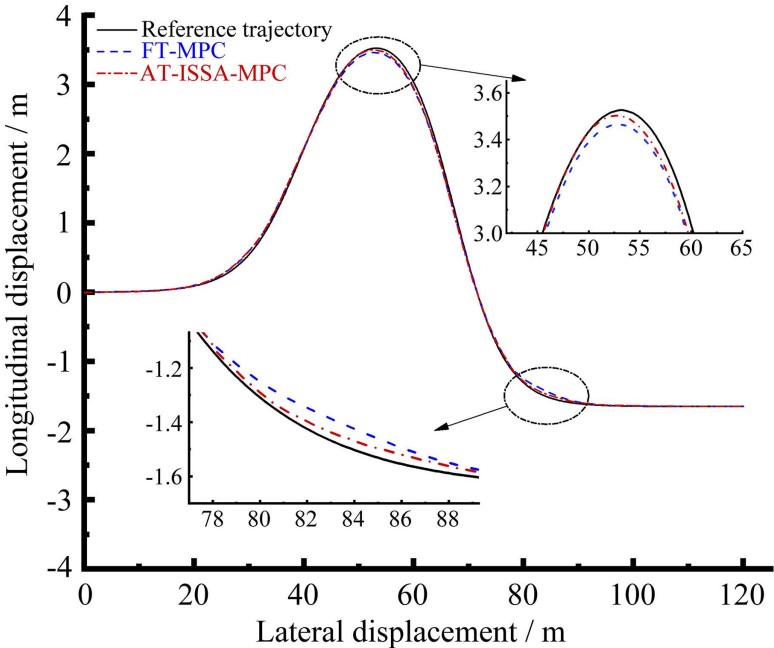

**Fig 6. Trajectory comparison.** AT-ISSA-MPC tracks the reference trajectory more closely than FT-MPC, achieving higher overall accuracy.

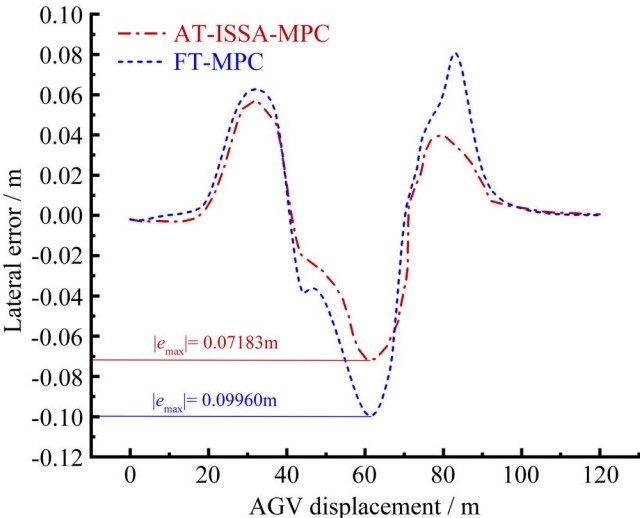

**Fig 7. Lateral error comparison.** AT-ISSA-MPC has smaller lateral error fluctuations, with a maximum error of 0.07183 m.

## Simulation analysis of the straight-line maneuver

The straight-line deviation correction scenario is designed to evaluate the controller's convergence performance under an initial lateral offset. The reference path is a straight line, and the vehicle starts with an initial lateral error of 0.5 m without an initial heading error.

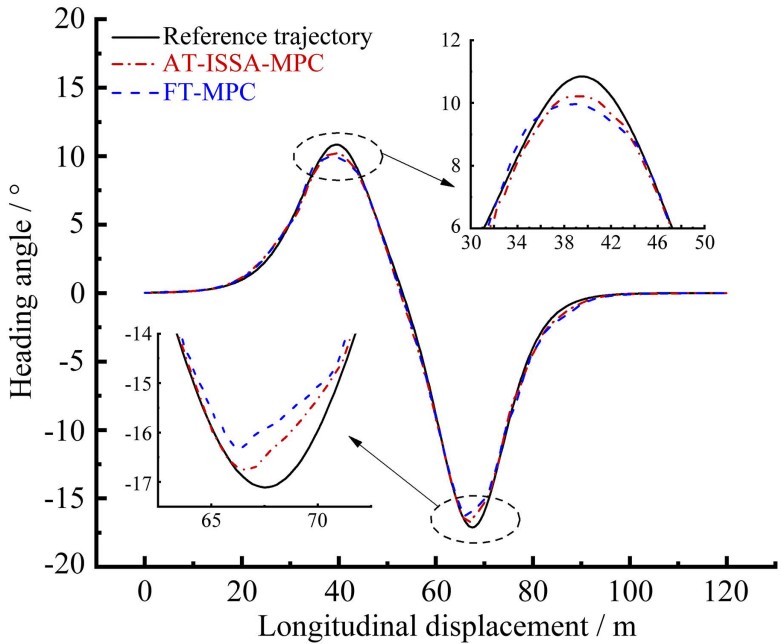

**Fig 8. Heading angle comparison.** AT-ISSA-MPC has smaller heading angle deviations (especially around curves), achieving higher overall heading tracking accuracy than FT-MPC.

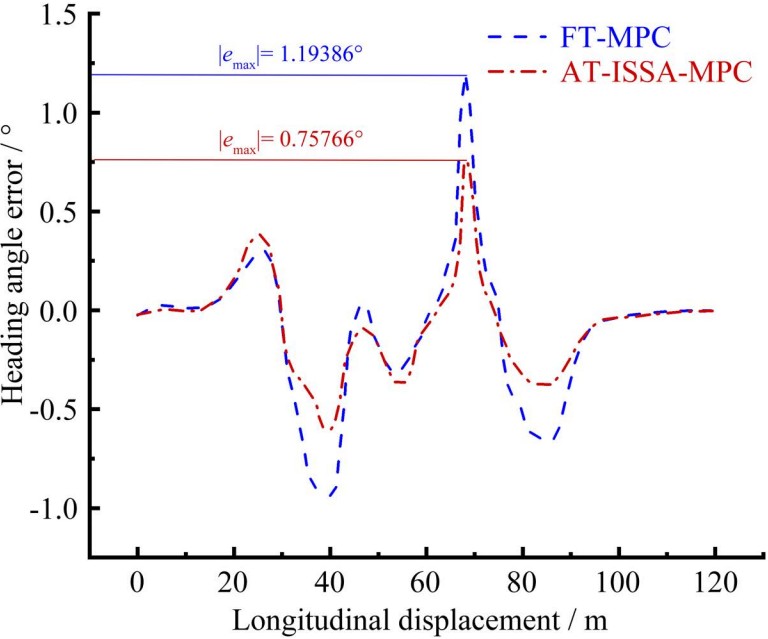

**Fig 9. Heading angle error comparison.** AT-ISSA-MPC has smaller heading angle error fluctuations, with a maximum error of 0.75766°.

**Table 1. Analysis of lateral displacement error.**

| Error metric | FT-MPC/ m | AT-ISSA-MPC/ m | Improvement/ % |
|---|---|---|---|
| Peak error | 0.09960 | 0.07183 | 27.88 |
| Mean square error | 0.001638 | 0.000795 | 51.45 |

**Table 2. Analysis of heading angle error.**

| Error metric | FT-MPC/ ° | AT-ISSA-MPC/ ° | Improvement/ % |
|---|---|---|---|
| Peak error | 1.19386 | 0.75766 | 36.54 |
| Mean square error | 0.118311 | 0.047652 | 59.72 |

Fig 10 presents the lateral error convergence curves of both algorithms under the same initial deviation. It can be clearly observed that the improved ISSA-MPC algorithm proposed in this paper converges to the vicinity of the reference trajectory within a shorter travel distance. In contrast, the classical MPC algorithm exhibits a longer convergence process. When the initial error is 0.5m, the improved ISSA-MPC is able to follow the intended trajectory after traveling 20m, which is 28.57% shorter than the 28m required by the classical MPC, indicating a significant improvement in convergence speed. A comprehensive analysis shows that the improved ISSA-MPC outperforms the classical MPC in terms of both tracking accuracy and stability, demonstrating its superior capability in enhancing tracking performance.

## Forklift-type AGV experimental validation

To validate the effectiveness of the proposed algorithm, real-vehicle experiments were conducted using a forklift-type AGV platform, as shown in Fig 11. The by-wire chassis is used to obtain relevant vehicle parameters, while a top-mounted

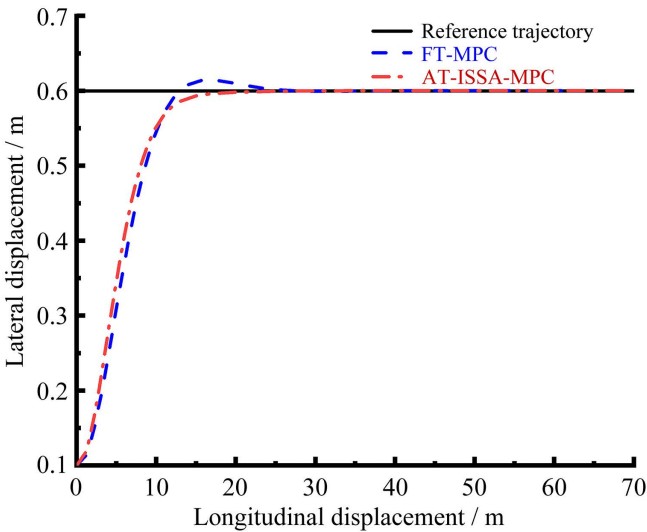

**Fig 10. Trajectory comparison.** AT-ISSA-MPC has smaller overshoot, faster convergence, and completes tracking earlier.To validate the performance of the proposed improved ISSA-MPC algorithm, a comparative experiment on straight-line path tracking was designed. Starting from the same initial positional deviation, both the improved ISSA-MPC algorithm and a classical MPC algorithm were tasked with tracking the same reference straight-line path.

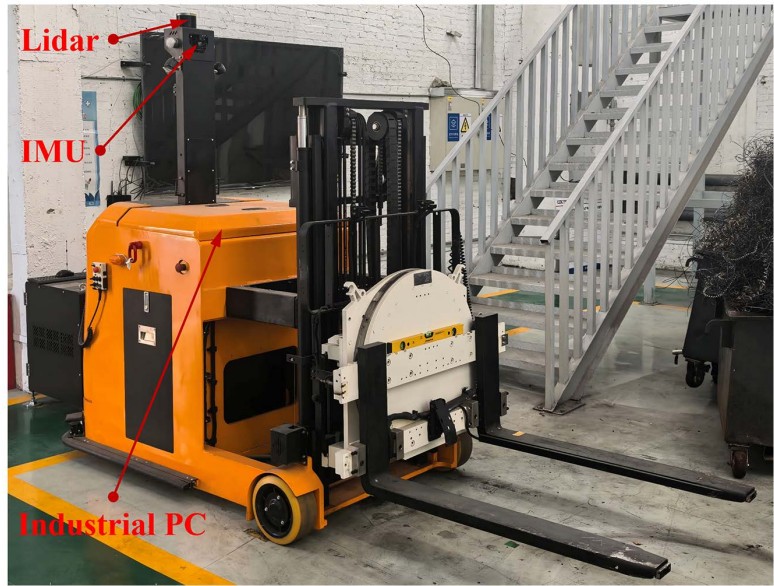

**Fig 11. Forklift-type AGV platform.** Figs 12 and 13 present a comparative analysis of the motion trajectories between the improved ISSA-MPC controller and the classical MPC controller on the experimental platform. Based on the data in Table 3, the following results are obtained: the maximum lateral displacement error of the classical MPC is 0.68502 m, while that of the improved ISSA-MPC is 0.34762 m, representing a reduction of 49.25% compared to the classical MPC. The mean square error of lateral displacement for the classical MPC is 0.069578 m, whereas that of the improved ISSA-MPC is 0.017335 m, a reduction of 75.09%. Based on the above analysis, under these test conditions, the improved algorithm proposed in this paper demonstrates satisfactory overall control performance in trajectory tracking, indicating good practical applicability.

LiDAR and an inertial measurement unit (IMU) system are responsible for collecting the vehicle's pose. In typical factory settings, the forklift-type AGV can travel at a speed of 1m/s. To ensure experimental safety, the speed was set to 0.5m/s for the 5m straight section before the turn, the curved section within the turn, and the 5m straight section after the turn, while the speed was maintained at 1m/s for other straight-line segments. The test scenario was an epoxy resin floor within a factory area.

In the real-vehicle experiments, the control cycle T was set to 40ms (25 Hz), consistent with the simulation configuration. The ISSA optimization was triggered every 50 control cycles (i.e., every 2s) as a background process. The measured average execution time of the AT-ISSA-MPC algorithm was 28.5ms, with a peak execution time of 35.6ms, both of which are well below the control cycle duration. This confirms that the selected T = 40ms provides sufficient margin against the peak computational load, achieving a practical balance between model fidelity and real-time performance. No control delays or stability issues were observed during operation, confirming that the proposed method satisfies real-time constraints on the embedded hardware platform, and the resulting tracking performance is shown in Fig 12 and 13.

In typical factory environments, AGVs often operate in narrow aisles with limited clearance between the vehicle and surrounding obstacles. As shown in Table 3, reducing the maximum lateral tracking error from 0.68502 m to 0.34762 m (a 49.25% reduction) significantly increases the safety margin against collisions, effectively mitigating the risk of path deviation-related incidents. Although both algorithms successfully guide the AGV to its destination, minimizing intermediate errors prevents error accumulation that could lead to positioning failures at critical points such as turns or docking stations. The consistency between these real-vehicle results and the simulation improvements (38–72% under double lane change) further confirms the practical relevance of the proposed method.

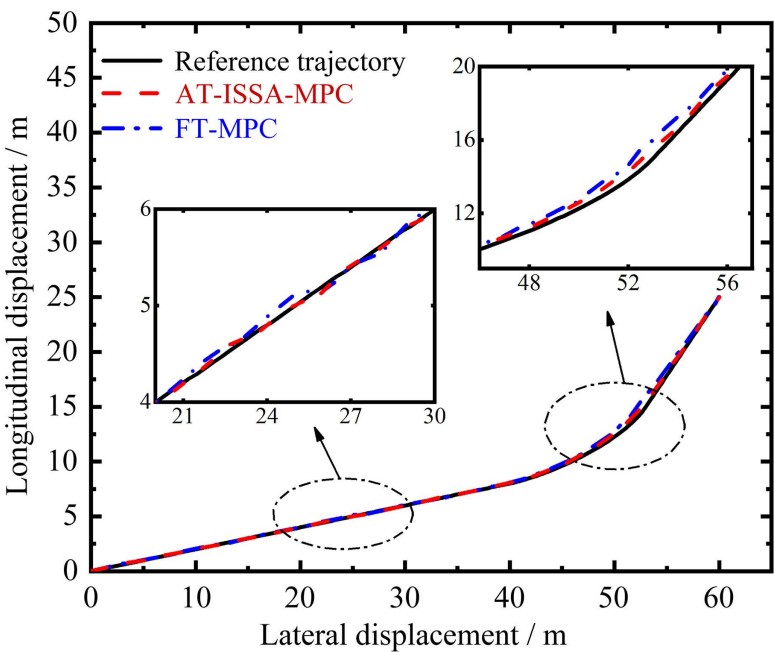

**Fig 12. Trajectory comparison.** AT-ISSA-MPC tracks the reference trajectory more closely with smaller overall fluctuations.

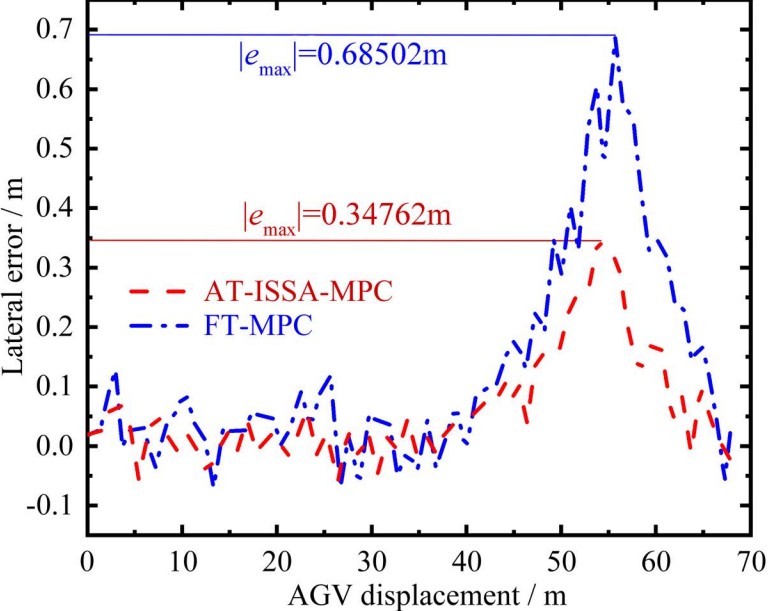

**Fig 13. Lateral error comparison.** AT-ISSA-MPC has smaller error fluctuations, higher accuracy, and better stability.

**Table 3. Analysis of lateral displacement error.**

| Error metric | FT-MPC/ m | AT-ISSA-MPC/ m | Improvement/ % |
|---|---|---|---|
| Peak error | 0.68502 | 0.34762 | 49.25 |
| Mean square error | 0.069578 | 0.017335 | 75.09 |

## Conclusion

This paper addresses the challenge of adaptively tuning the weight parameters in Model Predictive Control (MPC) for AGV trajectory tracking by proposing an online optimization strategy for MPC weights based on an Improved Sparrow Search Algorithm (ISSA). The strategy enhances the algorithm's global search capability and convergence stability by introducing Tent chaotic mapping for population initialization, incorporating a dynamic disturbance factor in the discoverer phase, and integrating a Cauchy mutation operator in the follower phase, effectively mitigating the tendency of traditional intelligent optimization methods to converge to local optima. Furthermore, a fitness function targeting comprehensive tracking error is constructed, enabling the periodic online optimization of MPC weight parameters. This allows the controller to adapt to the control requirements of the AGV under various operating conditions.

Through co-simulation using Gazebo and Rviz under the ROS framework, along with real-vehicle experiments, the effectiveness of the proposed adaptive time-domain MPC controller was validated. Under the double lane change maneuver, compared to the fixed-time-domain MPC, the ISSA-MPC reduced the peak lateral tracking error by 27.88% and its root mean square error (RMSE) by 51.45%. It also reduced the peak heading angle error by 36.54% and its RMSE by 59.72%. In the straight-line deviation correction scenario, the improved method achieved a 28.57% increase in convergence speed, demonstrating faster response and better stability. The real-vehicle tests further confirmed that the proposed algorithm maintains good tracking accuracy and practical applicability in real-world scenarios.

This research demonstrates that the MPC parameter adaptive optimization strategy based on the Improved Sparrow Search Algorithm can effectively enhance the trajectory tracking performance of AGVs in dynamically changing scenarios, providing a feasible solution for high-precision and highly adaptable control of AGVs. Future work will further investigate the coupled optimization problem in multi-AGV cooperative scheduling and explore the integration of reinforcement learning with MPC for more complex and uncertain environments.

In the experiments with the forklift-type AGV, the proposed ISSA-MPC controller demonstrated satisfactory tracking performance and exhibited good practical applicability. However, due to limitations such as localization accuracy, and the control precision and latency of the by-wire chassis, fluctuations in tracking error were observed under specific scenarios. Future research will focus on further optimization to address these issues.

## Acknowledgments

Thanks to Prof. Chengjun Ding for his guidance. Thanks to Tan Zhang, Jingyu Ge, Tengfei Ma, Zijian Li, Zhikai Jing, Jinshen Yu and Jianing Zhang for their assistance in the project.

## Author contributions

**Conceptualization:** Tan Zhang.

**Data curation:** Tengfei Ma.

**Formal analysis:** Tan Zhang.

**Funding acquisition:** Chengjun Ding.

**Investigation:** Zijian Li, Jingyu Ge.

**Methodology:** Chengjun Ding.

**Resources:** Jianing Zhang.

**Software:** Zhikai Jing, Jinshen Yu.

**Supervision:** Chengjun Ding, Ping Duan.

**Validation:** Zijian Li, Zhikai Jing, Jinshen Yu.

**Visualization:** Tengfei Ma.

**Writing – original draft:** Tan Zhang.

**Writing – review & editing:** Tan Zhang.

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
