## [Decision Letter · Decision Letter 0]

18 Mar 2026

PONE-D-26-10662Enhanced Accuracy and Adaptability: An ISSA-Optimized MPC Approach for AGV Trajectory TrackingPLOS One

Dear Dr. Ding,

Thank you for submitting your manuscript to PLOS ONE. After careful consideration, we feel that it has merit but does not fully meet PLOS ONE’s publication criteria as it currently stands. Therefore, we invite you to submit a revised version of the manuscript that addresses the points raised during the review process.

We look forward to receiving your revised manuscript.

Kind regards,

Jinhao Liang

Academic Editor

PLOS One

Journal Requirements:

2. Please note that PLOS One has specific guidelines on code sharing for submissions in which author-generated code underpins the findings in the manuscript. In these cases, we expect all author-generated code to be made available without restrictions upon publication of the work.

Please review our guidelines at https://journals.plos.org/plosone/s/materials-and-software-sharing#loc-sharing-code and ensure that your code is shared in a way that follows best practice and facilitates reproducibility and reuse.

“This work was supported by the Tianjin Science and Technology Plan Project (Grant No. 25ZXRGGX00310) and the Shijiazhuang Science and Technology Cooperation Special Project (Grant No. SJZZXA25003). Thanks to Prof. Chengjun Ding for his guidance. Thanks to Tan Zhang, Jingyu Ge, Tengfei Ma, Zijian Li, Zhikai Jing, Jinshen Yu and Jianing Zhang for their assistance in the project.”

5. We note that your Data Availability Statement is currently as follows:

“All relevant data are within the manuscript and its Supporting Information files.”

6. We note that Figure 11 in your submission contain copyrighted images. All PLOS content is published under the Creative Commons Attribution License (CC BY 4.0), which means that the manuscript, images, and Supporting Information files will be freely available online, and any third party is permitted to access, download, copy, distribute, and use these materials in any way, even commercially, with proper attribution. For more information, see our copyright guidelines: http://journals.plos.org/plosone/s/licenses-and-copyright.

1) You may seek permission from the original copyright holder of Figure 11 to publish the content specifically under the CC BY 4.0 license.

2) If you are unable to obtain permission from the original copyright holder to publish these figures under the CC BY 4.0 license or if the copyright holder’s requirements are incompatible with the CC BY 4.0 license, please either i) remove the figure or ii) supply a replacement figure that complies with the CC BY 4.0 license. Please check copyright information on all replacement figures and update the figure caption with source information.

If applicable, please specify in the figure caption text when a figure is similar but not identical to the original image and is therefore for illustrative purposes only.

7.If the reviewer comments include a recommendation to cite specific previously published works, please review and evaluate these publications to determine whether they are relevant and should be cited. There is no requirement to cite these works unless the editor has indicated otherwise.

Reviewers' comments:

Reviewer's Responses to Questions

**Comments to the Author**

1. Is the manuscript technically sound, and do the data support the conclusions?

Reviewer #1: Yes

Reviewer #2: Partly

2. Has the statistical analysis been performed appropriately and rigorously? 

Reviewer #1: Yes

Reviewer #2: No

3. Have the authors made all data underlying the findings in their manuscript fully available?

Reviewer #1: Yes

Reviewer #2: Yes

4. Is the manuscript presented in an intelligible fashion and written in standard English?

Reviewer #1: Yes

Reviewer #2: Yes

5. Review Comments to the Author

Reviewer #1: This manuscript proposes an adaptive trajectory tracking control framework for automated guided vehicles (AGVs) by integrating an Improved Sparrow Search Algorithm (ISSA) with Model Predictive Control (MPC) to optimize controller parameters online. The paper develops kinematic and dynamic models of a forklift-type AGV and validates the proposed strategy through simulation and real-vehicle implementation. The topic is relevant to intelligent logistics and AGV control, and the integration of optimization algorithms with MPC is a meaningful direction for improving adaptability. However, the manuscript still presents several limitations in terms of methodological clarity, theoretical justification, and depth of analysis. In its current form, the technical contributions and the advantages of the proposed approach are not sufficiently articulated. Therefore, I recommend major revision before the manuscript can be considered for publication.

1.The core contribution of the manuscript lies in improving the Sparrow Search Algorithm and using it to optimize MPC parameters online. However, the paper mainly introduces several heuristic modifications, including Tent chaotic initialization, disturbance factors, and Cauchy mutation, without clearly explaining the theoretical motivation behind these design choices. It would strengthen the manuscript if the authors could more clearly explain why these specific mechanisms improve the search behavior of the algorithm and how they address the limitations of the standard SSA in the context of MPC parameter tuning.

2.The paper states that ISSA is used to periodically optimize the weight parameters of the MPC cost function, but the integration between the optimization algorithm and the MPC controller is not fully clarified. In particular, the manuscript does not clearly describe how frequently the optimization is executed, how the optimization variables are defined, and how the updated parameters influence the control process over time. A clearer explanation of the interaction between ISSA and the MPC control loop would improve the transparency and reproducibility of the proposed framework.

3.The AGV dynamic model is derived based on several simplifying assumptions, such as small slip angles, planar motion, and simplified tire force relationships. While such assumptions are common in vehicle modeling, the manuscript does not sufficiently discuss their potential impact on control accuracy or the validity of the model under different operating conditions. A deeper discussion of the modeling limitations and their implications for the proposed control strategy would improve the technical rigor of the study.

4.The manuscript frequently claims that the improved algorithm enhances global exploration capability, prevents premature convergence, and improves tracking performance. However, these claims are primarily explained qualitatively. The paper would benefit from a more analytical discussion of how the algorithm modifications influence optimization behavior and why these changes are expected to produce better parameter tuning results for MPC.

5.Although the introduction reviews several trajectory tracking control approaches, the manuscript does not clearly position the proposed method within the broader research landscape of adaptive MPC and intelligent optimization-based controller tuning. Many existing studies have explored similar strategies that combine metaheuristic algorithms with MPC. The authors should more clearly highlight what differentiates their approach from existing methods and clarify the specific technical gap that this work aims to address.

6.The background should give a more detailed literature review on the path-tracking of the AGV. Some recent work, such as “A Polytopic Model-based Robust Predictive Control Scheme for Path Tracking of Autonomous Vehicles, IEEE Transactions on Intelligent Vehicles, vol. 9, no. 2, pp. 3928-3939, Feb. 2024” and “ETS-Based Human–Machine Robust Shared Control Design Considering the Network Delays, IEEE Transactions on Automation Science and Engineering, vol. 22, pp. 17501-17511, 2025” can be referred.

Reviewer #2: The authors presented an adaptive trajectory tracking strategy for automated guided vehicles (AGVs) by combining Model Predictive Control (MPC) with an Improved Sparrow Search Algorithm (ISSA). They established a kinematic and a simplified three-degree-of-freedom (3-DOF) dynamic model for a forklift-type AGV. To overcome the limitations of fixed-weight MPC, the authors propose tuning the time-domain parameters online using ISSA, which is enhanced via Tent chaotic mapping for population initialization, dynamic disturbance factors, and Cauchy mutation operators to balance global and local searches. The proposed AT-ISSA-MPC controller is validated against a standard Fixed Time-domain MPC (FT-MPC) through ROS/Gazebo simulations and a real-world test on a forklift AGV operating at 0.5m/s.

Decision: Major revision

Comments are attached separately.

6. PLOS authors have the option to publish the peer review history of their article (what does this mean?). If published, this will include your full peer review and any attached files.

Reviewer #1: No

Reviewer #2: No

---

## [Author Response · Author response to Decision Letter 1]

3 Apr 2026

We sincerely thank the editor and reviewers for their valuable feedback, which has helped us improve the quality of the manuscript. The comments from the editor and reviewers are listed below in bold and italicized font, and specific issues have been numbered. Our responses are provided in normal font, and the changes/additions to the manuscript are highlighted in blue.

As the paper template does not indicate section numbers, the corresponding section numbers are specified as follows:

0 Introduction

1 Establishment of the AGV motion model

1.1 Kinematic model

1.2 Dynamic model

2 MPC controller design

2.1 Linearization and discretization of nonlinear systems

2.2 Output prediction equations

2.3 Objective function design

2.4 Reformulation as a quadratic programming problem for solution

2.5 Feedback correction

3 Adaptive time-domain MPC controller design

3.1 Impact of time-domain parameters on tracking performance

3.2 Improved sparrow search algorithm

3.2.1 Discovery phase with dynamically adjusted exploration and exploitation

3.2.2 Adaptive mutation in the follower phase

3.2.3 Dynamic boundary constraints in the sentinel phase

3.3 Fitness function selection

3.4 Tent chaotic map

4 Simulation analysis and experimental verification

4.1 Simulation analysis of the double lane change maneuver

4.2 Simulation analysis of the straight-line maneuver

4.3 Forklift-type AGV experimental validation

5 Conclusion

The specific responses to the editor and reviewers' comments are as follows:

1. Journal Requirements

1.1 Response to Requirement 1

Requirement 1:

Please ensure that your manuscript meets PLOS ONE's style requirements, including those for file naming. The PLOS ONE style templates can be found at:

Response 1:

We sincerely thank the editor for the reminder. We have carefully reviewed the formatting requirements of PLOS ONE (including the two templates you provided: the main text formatting guide and the title page formatting guide) and will comprehensively revise the format, file naming, and other aspects of the manuscript to ensure full compliance with the journal’s style requirements.

Specifically, the revisions include: ①adding figure legends; ②adjusting the title format; ③increasing the line spacing of the main text; ④formatting the title, author names, and affiliations; ⑤removing funding information from the acknowledgments; ⑥revising the reference format; and ⑦deleting the Data Availability statement.

1.2 Response to Requirement 2

Requirement 2:

Please note that PLOS One has specific guidelines on code sharing for submissions in which author-generated code underpins the findings in the manuscript. In these cases, we expect all author-generated code to be made available without restrictions upon publication of the work.

Please review our guidelines at https://journals.plos.org/plosone/s/materials-and-software-sharing#loc-sharing-code and ensure that your code is shared in a way that follows best practice and facilitates reproducibility and reuse.

Response 2:

Thank you for your valuable suggestion regarding code sharing. We fully understand the importance of sharing code for enhancing research transparency and reproducibility. The source code for this study was developed within a technology transfer project in collaboration with an enterprise, and its ownership is jointly held by both parties. After consultation with the enterprise, due to intellectual property considerations, including pending patents, they have requested that the code not be publicly released at this stage. In light of these confidentiality requirements, we are currently unable to make the code publicly available. However, once the intellectual property issues are resolved, we would be happy to consider reasonable requests for access to the code for the purpose of verifying the results of this study.

1.3 Response to Requirement 3

Requirement 3:

We note that the grant information you provided in the ‘Funding Information’ and ‘Financial Disclosure’ sections do not match.

Response 3:

Thank you for pointing out the inconsistency in our funding information. We have carefully reviewed and corrected the details.

The correct information for both sections is as follows:

Funding information (in the submission system):

Funder 1: Tianjin Science and Technology Plan Project

Grant number 1: 25ZXRGGX00310

Funder 2: Shijiazhuang Science and Technology Cooperation Special Project

Grant number 2: SJZZXA25003

Financial Disclosure statement: “This work was supported by the Tianjin Science and Technology Plan Project (Grant No. 25ZXRGGX00310) and the Shijiazhuang Science and Technology Cooperation Special Project (Grant No. SJZZXA25003).”

Since we could not find the Financial Disclosure option in the submission system, we kindly request that you update the information on our behalf.

Please let us know if any further adjustments are needed.

1.4 Response to Requirement 4

Requirement 4:

Thank you for stating the following in the Acknowledgments Section of your manuscript:

“This work was supported by the Tianjin Science and Technology Plan Project (Grant No. 25ZXRGGX00310) and the Shijiazhuang Science and Technology Cooperation Special Project (Grant No. SJZZXA25003). Thanks to Prof. Chengjun Ding for his guidance. Thanks to Tan Zhang, Jingyu Ge, Tengfei Ma, Zijian Li, Zhikai Jing, Jinshen Yu and Jianing Zhang for their assistance in the project.”

Response 4:

Thank you for your guidance regarding the placement of funding information. We have revised the manuscript accordingly and would like to update the Funding Statement as follows.

(1) Removal from the Acknowledgments section

We have removed the following funding-related text from the Acknowledgments section of the manuscript:

“This work was supported by the Tianjin Science and Technology Plan Project (Grant No. 25ZXRGGX00310) and the Shijiazhuang Science and Technology Cooperation Special Project (Grant No. SJZZXA25003).”

The Acknowledgments section now contains only personal acknowledgments, as follows:

“Thanks to Prof. Chengjun Ding for his guidance. Thanks to Tan Zhang, Jingyu Ge, Tengfei Ma, Zijian Li, Zhikai Jing, Jinshen Yu and Jianing Zhang for their assistance in the project.”

(2) Updated Funding Statement

Please replace the current Funding Statement (“The authors received no specific funding for this work.”) with:

“This work was supported by the Tianjin Science and Technology Plan Project (Grant No. 25ZXRGGX00310) and the Shijiazhuang Science and Technology Cooperation Special Project (Grant No. SJZZXA25003).”

We have ensured that no funding information appears elsewhere in the manuscript. As we could not locate the Funding Statement option in the submission system, we kindly request that you update it on our behalf.

Please let us know if any further adjustments are needed.

1.5 Response to Requirement 5

Requirement 5:

We note that your Data Availability Statement is currently as follows:

“All relevant data are within the manuscript and its Supporting Information files.”

Response 5:

Thank you for your inquiry regarding our Data Availability Statement. We have carefully reviewed PLOS’s policy on the minimum dataset.

Regarding shareable data: We have provided all processed data necessary to generate the figures and statistical analyses presented in the manuscript. These data, including the trajectories in the path tracking plots and the error metrics reported in the results, are available in the Supporting Information file (Figures_and_Related_Data.zip). This dataset is sufficient to verify the comparative performance conclusions presented in this study.

Regarding restricted raw data: The complete raw data logs (e.g., high-frequency sensor data streams, internal control signals) are owned by our industrial partner as part of a technology transfer agreement and are subject to a pending patent. Due to these intellectual property and confidentiality considerations, the full raw dataset cannot be made publicly available.

Updated Data Availability Statement: We have revised the Data Availability Statement to read as follows:

“No - some restrictions will apply.”

The TIFF figures and related data have been uploaded to the submission system under the file name: Figures_and_Related_Data.zip.

We believe this approach satisfies the journal’s requirements while respecting the necessary intellectual property protections.

1.6 Response to Requirement 6

Requirement 6:

We note that Figure 11 in your submission contain copyrighted images. All PLOS content is published under the Creative Commons Attribution License (CC BY 4.0), which means that the manuscript, images, and Supporting Information files will be freely available online, and any third party is permitted to access, download, copy, distribute, and use these materials in any way, even commercially, with proper attribution. For more information, see our copyright guidelines: http://journals.plos.org/plosone/s/licenses-and-copyright.

1) You may seek permission from the original copyright holder of Figure 11 to publish the content specifically under the CC BY 4.0 license.

2) If you are unable to obtain permission from the original copyright holder to publish these figures under the CC BY 4.0 license or if the copyright holder’s requirements are incompatible with the CC BY 4.0 license, please either i) remove the figure or ii) supply a replacement figure that complies with the CC BY 4.0 license. Please check copyright information on all replacement figures and update the figure caption with source information.

If applicable, please specify in the figure caption text when a figure is similar but not identical to the original image and is therefore for illustrative purposes only.

Response 6:

Thank you for your careful review of Fig. 11. We would like to clarify that this figure is entirely original and created by the authors.

The forklift-type AGV shown in Fig. 11 is a self-developed prototype modified from a traditional electric forklift. The photograph was taken by the authors, and all annotations were manually added. No third-party copyrighted materials are included in this figure.

Therefore, this figure fully complies with the CC BY 4.0 license requirements, and no external copyright permission is needed.

Please let us know if any further documentation or confirmation is required.

1.7 Response to Requirement 7

Requirement 7:

Response 7:

Thank you for your attention. We have evaluated the literature recommended by the reviewer and related studies, and carefully reviewed the suggested references, confirming that they are highly relevant to this work. Therefore, we have incorporated them into the revised manuscript.

Specifically, the following references have been added to the Introduction section to support the literature review on adaptive MPC and trajectory tracking control (Page 2):

25. Jinhao Liang, Qingyun Tian, Jiwei Feng, Dawei Pi, Guodong Yin. A Polytopic Model-Based Robust Predictive Control Scheme for Path Tracking of Autonomous Vehicles. IEEE Transactions on Intelligent Vehicles. 2024; 9(2): 3928-3939. DOI: 10.1109/TIV.2023.3340668

26. Jinhao Liang, Yanbo Lu, Faan Wang, Jiwei Feng, Dawei Pi, Guodong Yin. ETS-Based Human–Machine Robust Shared Control Design Considering the Network Delays. IEEE Transactions on Automation Science and Engineering. 2025; 22: 17501-17511. DOI: 10.1109/TASE.2024.3383094

35. Qi Xia, Peng Chen, Guoyan Xu, Bin Zhou, Guizhen Yu. Reinforcement Learning-Based Adaptive Safety Tracking Controller for Autonomous Mining Trucks. IEEE Transactions on Vehicular Technology. 2025; 74(7): 10122-10136. DOI: 10.1109/TVT.2025.3546647

36. Kairui Chen, Yixiang Gu, Weicong Huang, Zhonglin Zhang, Zian Wang, Xiaofeng Wang. Fixed-Time Adaptive Event-Triggered Guaranteed Performance Tracking Control of Nonholonomic Mobile Robots under Asymmetric State Constraints. Mathematics. 2024; 12(10): 1471. https://doi.org/10.3390/math12101471

37. Huarong Zheng, Rudy R. Negenborn, Gabriël Lodewijks. Fast ADMM for Distributed Model Predictive Control of Cooperative Waterborne AGVs. IEEE Transactions on Control Systems Technology. 2017; 25(4): 1406-1413. DOI: 10.1109/T

---

## [Decision Letter · Decision Letter 1]

4 May 2026

Enhanced accuracy and adaptability: an ISSA-optimized MPC approach for AGV trajectory tracking

PONE-D-26-10662R1

Dear Dr. Ding,

We’re pleased to inform you that your manuscript has been judged scientifically suitable for publication and will be formally accepted for publication once it meets all outstanding technical requirements.

Kind regards,

Jinhao Liang

Academic Editor

PLOS One

Additional Editor Comments (optional):

Reviewers' comments:

Reviewer's Responses to Questions

**Comments to the Author**

1. If the authors have adequately addressed your comments raised in a previous round of review and you feel that this manuscript is now acceptable for publication, you may indicate that here to bypass the “Comments to the Author” section, enter your conflict of interest statement in the “Confidential to Editor” section, and submit your "Accept" recommendation.

Reviewer #1: All comments have been addressed

Reviewer #2: All comments have been addressed

2. Is the manuscript technically sound, and do the data support the conclusions?

Reviewer #1: Yes

Reviewer #2: Yes

3. Has the statistical analysis been performed appropriately and rigorously? 

Reviewer #1: Yes

Reviewer #2: Yes

4. Have the authors made all data underlying the findings in their manuscript fully available?

Reviewer #1: Yes

Reviewer #2: Yes

5. Is the manuscript presented in an intelligible fashion and written in standard English?

Reviewer #1: Yes

Reviewer #2: Yes

6. Review Comments to the Author

Reviewer #1: This manuscript proposes an adaptive trajectory tracking control framework for automated guided vehicles (AGVs) by integrating an Improved Sparrow Search Algorithm (ISSA) with Model Predictive Control (MPC) to optimize controller parameters online.

Reviewer #2: The authors have systematically addressed every major and minor concern raised, significantly elevating the scientific rigor, reproducibility, and practical relevance of the manuscript.

Specifically, the following revisions have greatly strengthened the paper:

Computational Feasibility: By explicitly providing the control cycle (40 ms) and the measured execution times of the AT-ISSA-MPC algorithm (28.5 ms average, 35.6 ms peak), the authors have completely resolved my concerns regarding the real-time viability of running an online metaheuristic optimizer. The explanation of the decoupled 2-second background trigger was an excellent clarification.

Modeling Assumptions: The addition of quantitative boundaries for the dynamic model—specifically capping the operational speed at 1.5 m/s and limiting the linear tire model assumption to slip angles of 3-5 degrees—successfully grounds the theoretical control design in physical, industrial reality.

Control-Theoretic Depth: The updated analysis of the double lane change and straight-line deviation scenarios is excellent. The explicit explanation of how the ISSA algorithm dynamically adjusts the prediction (N_p) and control (N_c) horizons to anticipate curves and suppress transient overshoot provides the exact mechanistic depth that was previously missing.

Practical Significance: The authors successfully connected their numerical error reductions to real-world industrial constraints. Framing the 49% reduction in peak lateral error in the context of narrow-aisle collision avoidance and precision docking effectively answers the "practical significance" critique.

Literature and Citations: The expanded introduction, which now appropriately contextualizes the ISSA approach against modern trends like Reinforcement Learning and ADMM, provides a much stronger theoretical framing for the paper. Additionally, providing concrete citations for the limitations of "traditional methods" reflects sound academic practice.

7. PLOS authors have the option to publish the peer review history of their article (what does this mean?). If published, this will include your full peer review and any attached files.

Reviewer #1: No

Reviewer #2: No

---

## [Editor Report · Acceptance letter]

PONE-D-26-10662R1

PLOS One

Dear Dr. Ding,

I'm pleased to inform you that your manuscript has been deemed suitable for publication in PLOS One. Congratulations! Your manuscript is now being handed over to our production team.

Kind regards,

on behalf of

Dr. Jinhao Liang

Academic Editor

PLOS One